# Complexity in Geophysical Time Series of Strain/Fracture at Laboratory and Large Dam Scales: Review

**DOI:** 10.3390/e25030467

**Published:** 2023-03-07

**Authors:** Tamaz Chelidze, Teimuraz Matcharashvili, Ekaterine Mepharidze, Nadezhda Dovgal

**Affiliations:** Mikheil Nodia Institute of Geophysics, Ivane Javakhishvili Tbilisi State University, Tbilisi 0179, Georgia

**Keywords:** geophysical complexity, complexity measures, stick–slip complexity, synchronization by weak forcing, large dam monitoring, nonlinear dynamics of strain/seismicity near large dam

## Abstract

One of the interesting directions of complexity theory is the investigation of the synchronization of mechanical behavior of large-scale systems by weak forcing, which is one of manifestations of nonlinearity/complexity of a system. The effect of periodic weak mechanical or electromagnetic forcing leading to synchronization was studied on the laboratory load–spring system as well as on a big dam’s strain data. Due to synchronization, the phase space structure of the forced system strongly depends on the weak forcing intensity–determinism show itself in the recurrence of definite states of the forced system. The nonlinear dynamics of tilts/strains/seismicity near grand dams reflect both the complexity of the mentioned time series, connected with the natural agents (regional and local geodynamics), which were presented even before dam erection, as well as the effects of the water level (WL) variation in the reservoir, which is a quasi-periodic forcing superimposed on the natural geodynamic background. Both these effects are documented by the almost half-century of observations at the large Enguri Dam. The obtained data on the dynamics of strain/seismicity near a large dam can be used for the assessment of the possible risks, connected with the abrupt change of routine dynamics of construction.

## 1. Introduction

As Stephen Hawking wrote, the 21st century will be the century of complexity. It seems that the prediction is fulfilled. As a rule, the complexity of the strain/fracture process of real systems makes it impossible to study their behavior in the frame of classic linear theories. Complex systems reveal the property of “emergency”, which means that their elements interact in a collective way, leading to a phenomenon of “self-organization” and to the appearance of structures with non-integer-dimension fractals. Approximately four decades ago the percolation model of complex (multiple) fracture of solids and seismic process was suggested [1,2,3]. The critical crack concentration, corresponding to the formation of the infinite percolation cluster of fractures, was considered as the criterion of macroscopic failure of a system. Chelidze ([3], submitted for publication in 1979) considered the physical mechanism of dispersed fracture of solids, corresponding to the percolation criticality model. The volume concentration of cracks was suggested as the order parameter of the critical process of multiple fractures. Several months later, [4] published a paper where a similar approach to the multiple process of fragmentation of solids was developed. The classic percolation approach to the fracture process consists of modeling the destruction of the solid beginning from the nucleation of separate elementary cracks, to their multiplication/coalescence leading to clustering and the formation of an infinite cluster [3]. Its application to the cyclic seismic process should include the model of repeated fracture of a system, where the seismic cycle is considered as a process with intermittent criticality, i.e., approaching to and retreat from the critical state of a fault network. This means that after dynamical stress drop, the internal friction of faults increases with time due to the healing action of high temperatures and confining pressures, giving start to a new stress build-up process [5].The seismic process is also a complex one: the Earth’s core and mantle are composed of many components, interacting nonlinearly; correspondingly, the behavior of such systems in time and space is a subject of nonlinear dynamics. This approach, which is rooted in the works of Poincare and Lyapunov, requires a detailed knowledge of the real seismogenesis process, which can be described by a system of differential equations [6,7]. Using modern methods of complexity analysis reveals the existence of long-term correlations in the temporal, spatial and energy distributions in dynamical systems, such as the fracture of solids and seismic processes [8,9,10,11].

At present, the modern machine learning (ML) approach, which is concerned with developing algorithms and improving their performance with an increasing volume of input information [12], has gained increasing attention in solving problems where it is impossible to formulate exact mathematical models, though on the other hand there are many real data measurements. ML allows the creation of a data-driven approach to understanding and forecasting the behavior of many complex systems without constructing the exact analytical model, in contrast to complexity theory. The connection between machine learning and nonlinear dynamics/complexity theory at present is still unexplored and is the object of a future special issue of Physica D planned for June 2023 (https://www.sciencedirect.com/journal/physica-d-nonlinear-phenomena/about/call-for-papers (accessed on 23 January 2023)). 

The effect of synchronization of a large-scale system’s mechanical behavior with a weak forcing is one of the manifestations of nonlinearity/complexity of a system [7,13,14]. Due to synchronization, the phase space structure of the forced system depends strongly on even the weakest forcing intensity/frequency—the existence of determinism show itself in the recurrence of definite states of the forced system. Note that besides the 1:1 phase locking at the applied frequency ω to a natural one $\omega_{0}$ of the process ($\omega \approx \omega_{0}$), there is also the high-order synchronization at the natural frequency multiples ($m\omega_{0}$): $\omega =m\omega_{0}$, where m is the winding number [8,14]. There are several methods for the quantitative analysis of the synchronization strength between a forced process and forcing signal, namely: compilation of the Arnold’s plots (phase space plot of forcing intensity versus forcing frequency); calculation of the generalized phase difference; Shannon and Tsallis entropies; conditional probability of phases; flatness of stripes of synchrograms (stroboscopic approach); small-word connectivity and the coefficients of phase diffusion [8,13,14,15]. 

As revealed during recent years, a reaction of heterogeneous materials such as rocks, concrete, etc. to the applied mechanical impact is a complex one and significantly differs from the behavior of a system’s homogeneous components. The corresponding tool for the interpretation of experimental stress–strain diagrams is suggested by the theory of mesoscopic elasticity [16], which is an important complement to the classic mechanics of heterogeneous materials. The stress–strain (or tilt) dependencies of such systems manifest nonlinear hysteretic behavior, namely, the asymmetric response to loading and unloading, due to so-called mesoscopic structural features (a system of microcracks). Heterogeneous materials contain an enormous number ($10^{9}-10^{12}$) of such defects per square centimeter, which means that the macroscopic elastic properties of a material depend strongly on the behavior of microcracks. Thus, the characteristics of the hysteretic cycle can be used for diagnostics of a material: in the absence of cracks, the brittle solid manifests linear elasticity without any hysteresis, and the appearance of cracks leads to hysteresis; the opening of the hysteresis curve increases with the number of defects. The approximation of hysteretic data can be accomplished using the Preisach–Mayergoyz (P-M) phenomenological model in which the system is represented by a complex of hysteretic elastic units or hysterons, manifesting an asymmetric response to load–unload cycles [16]. 

## 2. Materials and Methods

### Complexity Measures for Interpretation of Geophysical Monitoring Data

Geophysical systems/processes are complex due to their nonlinearity [6]. The concept of complexity is related to phenomena with a broad diversity of dynamical features, from strict determinism to total randomness; in between, there are intermediate states with different levels of complexity: quasiperiodicity, deterministic chaos, etc. [17]. For measuring the degree of complexity of a given process, Takens [18] suggested the time-delay reconstruction method of the original time series. At present, the popular tools for qualitative/quantitative complexity analysis are recurrence plots (RP) and recurrence quantitative analysis (RQA) [19], mutual information (MI) [20], the Lempel–Ziv (algorithmic) complexity measure (LZC) [21,22], singular spectrum analysis (SSA) [23], the Bolzmann–Gibbs–Shannon and Tsallis entropies [24], the Mahalanobis distance (MD) [25], and the natural time analysis approach [26,27,28].

Nonlinear/complex systems under oscillating weak impact quite frequently manifest the effect of adjustment of rhythms or synchronization [7,13,14]. Some of nonlinear dynamics tools can be used for revealing synchronization and measuring its strength. In addition, there are some effective methods for measuring the synchronization strength between a forcing signal and forced system: phase space plots of forcing frequency versus forcing intensity or Arnold’s plots [29], detrended fluctuation analysis (DFA) [30], generalized phase difference, the stroboscopic approach and coefficients of phase diffusion [13,14]. The mentioned methods are very useful for the analysis of integrate and fire (relaxation) processes—the multiple fractures belong to this class. There are specific methods for the analysis of noisy and short time series—singular spectral analysis (SSA) [31,32] as well as the simple Schuster test [33].

In recent years, it has been shown that many complex systems do not always obey the classic Boltzmann–Gibbs–Shannon approach; in such systems, the probability of occurrence of different microstates is correlated: the distribution of events in such systems obey a power law due to long-range interactions. Tsallis [24] suggested a generalized approach, non-extensive statistical mechanics (NESM), in which interactions among the elements of a system at all lengths are taken into account. It has been shown in many publications that the Tsallis entropy helps to build a new statistical mechanics, which is very useful for the description of fractures from the laboratory scale to a global seismic process [34,35,36]. Tsallis entropy calculation is an often-used method of different complex time-series analysis. Usually, subsystems of the real-world systems are not independent and thus the Boltzmann–Gibbs–Shannon condition of extensivity is not satisfied. Consequently, long- and short-range correlations in such systems are far from able to be regarded as negligible at all scales. Thus, we face a nonextensive case for which Tsallis [24] introduced a special entropic expression:(1)$$S_{T}=\frac{k}{q-1}\left(1-\sum_{i=1}^{\Omega }p_{i}^{q}\right)$$
where *k* is Boltzmann’s constant and Ω is the total number of accessible *i* microstates of the system with a probability of occupation $p_{i}$. The number *q*, which characterizes the degree of non-extensivity of the system, is an entropic index. The Tsallis entropy, which quantifies the dynamic changes in the complexity of the system, is lower for the cases that are characterized by lower complexity (are less random).

As the entropy (in reality, the entropy change) is the amount of disorder generated by some impact on the system, a negative entropy means there is less disorder after an impact to the system. In our experiments we organize/increase the order in the system by applying some weak periodical forcing, which leads to a synchronization effect. For example, in the stick–slip experiments in the slider–spring systems without external forcing, the distribution of the waiting times between slips is wide; after the addition of a weak periodic forcing to the main spring pull the disorder in the waiting times between slips decrease strongly [8]. The corresponding entropy in the waiting time distribution during forced stick–slip greatly decreases compared to the experiments without periodic forcing due to the increased ordering in the waiting time distribution (entropy decreases and even can become negative). Of course, the effect of ordering (appearance of negative entropy in the waiting time distribution) is due to energy spent (locally) for the organization of the process (in our case, by the generator of periodic forcing, or periodic water load/unload in the reservoir). At the same time, according to the second law of thermodynamics, a system as a whole always moves towards disorder due to the energy spent for process organization.

## 3. Complexity in Acoustic Time Series during Laboratory Stick–Slip

Stick–slip or non-stable friction, after the basic works of Brace and Byerlee [37] and Ruina [38], is considered as a foundation of the modern concept of seismic processes. Acoustic emission (AE), which accompanies the unstable friction, allows us to study the dynamics of this complex process. As in many complex phenomena, stick–slip motion in some specific conditions reveals an anomalously high sensitivity to a weak external impact, namely, synchronization by small forcing [8]. In principle, a percolation process can be considered as a model of elementary slip at a tangential displacement of contacting fractal surfaces at a critical number of contacts between them [3]. There are experimental data indicating that before tangential displacement the contact points number *n* between fractal surfaces decreases gradually due to their separation until attaining the critical value $n_{c}$, at which the tangential force exceeds the friction force. Ben-David et al. [35] confirm that there is really some critical value of contacts’ number drop—20%—after which the slipping of surfaces occurs due to expulsion of the sliding plates. This percolation model of elementary slip can be developed for the dynamic stick–slip process: according to [39], during stick–slip the dynamics of the system undergoes several phases: first, a strong reduction in the contact area (until attainment of the percolation threshold), then the fast-slip phase, followed by a slow slip, after which the system returns to a stick state. The initial contact area is restored due to the immediate closure of the elementary cracks after the final rupture (strain release) of the solid or due to ageing after some characteristic time. Experimental acoustic emission (AE) systems developed in recent years allow us to obtain detailed data on the microphysics of stick–slip processes, including important details on the precursory events signaling that the system is close to the critical (macro-slip) phase [40,41].

### Periodically Forced Stick–Slip

The slider–spring set is a simple kind of relaxation oscillator that in certain conditions produces quasiperiodic stick–slip motion. If, besides the linearly increasing pull of the spring, a weak periodic mechanical/electric stress is applied to the sample, the summary stress $\tau_{s}$ is:(2)$$\tau_{s}(t)=\tau (t)+a\sin(\omega t+\phi )$$
where a, ω, ϕ are the amplitude, frequency and initial phase, respectively. At the critical value of shear stress $\tau_{s}=\tau_{c}$, the spring pull overcomes the friction resistance of the slider and the slider slips on same characteristic distance:(3)$$\tau_{s}(t)=\tau (t)+a\sin(\omega t+\phi )=\tau_{c}$$

The stick–slip process due to its nonlinear nature can be strongly affected by a weak impact. In particular, the spring–slider system, which is an example of an autonomous oscillator with a natural frequency $\omega_{0}$, can be affected by a weak periodic forcing of frequency ω and intensity *I*: it change its frequency to some resulting value Ω [16,17], which can differ from $\omega_{0}$. This effect is the result of the so-called phase synchronization (PS), when the frequencies ω and Ω are adjusted, but the amplitudes of the oscillating system can be irregular. The value ($\omega -\omega_{0}$) or detuning is minimal when $\omega \approx \omega_{0}$. As a result of synchronization the plot of *I* versus ω looks like an inverse bell curve with a minimum at ($\omega =\omega_{0}$); the curve is called Arnolds’ tongue [29]. The so-called high order synchronization can also occur at frequencies $m\omega_{o}$ [14].

The above approach to the triggered stick–slip can be applied to the interpretation of induced/triggered seismicity. Seismicity can be triggered by very weak forcing, such as hydrocarbon or geothermal energy production, wave trains of strong remote earthquakes (dynamic tremors), the influence of Earth’s tides, reservoir impoundment and underground fluid and gas storages [42,43,44]. According to above publications, the EQ activity can be modulated/synchronized by a mechanical impact on the order of a few thousand KPa/hundreds of Pa; such a strain can be induced by a water table change on the order of dozens of cm. These additional impacts are much less than the main mechanical stresses at depths of 10–20 km, where the lithostatic stress of the upper layers is on the order of 100 MPa.

## 4. Results

### 4.1. Nonlinear Dynamics of Stick–Slip at Mechanical Forcing

Figure 1 presents experiments on the phase synchronization of the nonlinear spring–slider system by a weak periodic mechanical forcing applied either normal or parallel to the slip surface [45,46]. The maximal intensity of mechanical forcing $F_{max}$ at a maximal applied voltage of 6.5 V was on the order of $2*10^{-3}\;N$, which is much less than the driving force of the pulling spring, 4 N. For smaller applied voltages the forcing is even weaker.

The registered AE data presented in Figure 1 correspond to the slip events triggered by mechanical forcing. It is evident that at low forcing voltages at both normal and tangential forcing the AE onsets are registered in all decimals of the forcing periods in a more or less similar way. At the higher forcing voltages, emissions concentrate at the definite parts of the forcing period, which point to an increasing synchronization of slip events by the weak impact: the slips occur only at the definite forcing phases, i.e., the application of weak mechanical forcing, a thousand times weaker than the main driving force of the spring, promotes the phenomenon of synchronization.

### 4.2. Nonlinear Dynamics of Stick–Slip at Electromagnetic Forcing

Experiments on the synchronization of stick–slip by periodic electromagnetic (EM) forcing were carried out on the system of two plates of basalt: to the upper plate, which was sliding under the pull (10 N) of a spring, was applied an additional weak periodic electromagnetic forcing of variable frequency and amplitude (from 0 to 1000 V) normal to the sliding plane. The mechanical action of such forcing was much weaker (equivalent to ≈1N) compared to the pulling force [33]. The system without forcing followed the typical chaotic stick–slip pattern for such systems, but after the application of forcing it became completely phase-synchronized (Figure 2).

An EM field applied to the dielectric generates in it the additional electrostriction force $F_{sum}$, which pushes the sample in the direction of increasing intensity; this force $F_{sum}=\mu \left(F_{rs}+F_{pi}\right)$, where $F_{rs}$ is the friction force and $F_{pi}$ is the additional friction due to the applied EM force. The above expression is similar to Hubbert and Rubbey’s [47] (1959) formula for the friction force in the slip surface between two samples of porous solids taking into account the pore pressure of the fluid. The electro-striction force $F_{pi}$ is the analogue of the pore pressure term in Hubbert and Rubbey’s model. The EM synchronization of stick–slip was observed only at the special values of characteristic parameters: applied forcing frequency *f*, stiffness of spring $K_{s}$ and applied voltage $V_{a}$ [45].

Various tools of nonlinear dynamics/synchronization theory [13,14,17,48,49] were applied to the experimental data of stick–slip at a variable intensity of forcing (Figure 3, Figure 4 and Figure 5) [8,45,46]; namely, the waiting time ordering under sinusoidal forcing synchronization area (Arnold’s tongue) of stick–slip for the mentioned set of characteristic parameters was found (Figure 5), as well as the phase diffusion coefficient of phase differences between AE maxima and the forcing phase, Shannon entropy (Figure 4), mutual information (MI) of phase differences in the consecutive time windows, etc.

In Figure 3 we present time series of waiting times between the amplitudes of acoustic signals in the consecutive Δt periods of external forcing for a whole record. Significant EM forcing was applied in the time interval from 22 to 60 s of the stick–slip process; in this interval the scatter of waiting times decreases significantly.

Figure 4 presents the variation of the Shannon entropy value *S*: the clear decrease in *S* indicates that the dynamics of acoustic emission becomes much more regular in the synchronized part of the acoustic emission data set, from 22 to 60 s.

The “phase diagram” for variables *f* and $V_{a}$, or the so-called Arnold’s tongue [29], is presented in Figure 5. It is evident that phase space (Arnold’s tongue) plots at external forcing have a bell-curve form. The minimum of the forcing voltage is observed at 60 Hz.

All these experiments give similar results, namely, a strong increase in the ordering of time intervals between sequential stick–slip events Δt in AE time series, which is a signature of stick–slip synchronization by EM forcing.

### 4.3. Complexity in Reservoir Induced Tilts/Strains/Seismicity

#### 4.3.1. Enguri Dam System

Nonlinear dynamics of tilts/strains/seismicity near grand dams reflects both the features of the mentioned time series, connected with the natural agents (regional and local geodynamics, which were present even before dam erection), as well as the effects of the water level variation in the reservoir, which is a quasi-periodic forcing superimposed on the natural geodynamic background. Both these effects are documented by the observation system at the large Enguri Dam (Figure 6). The Enguri Dam (Western Georgia, 42.030 N, 42.775 E) has a height of 272 m; the volume of water in its reservoir is 1.1 × 10 ^9^ m^3^. Filling of the lake started in December 1977; the water reached the maximal height in 1986. The monitoring system of the Enguri Dam area has been functioning from 1971, well before lake filling, which began in 1978. After 1986, the reservoir was filled and emptied in a regular manner, almost periodically. Due to the almost half-century of monitoring, a long time series of deformation/tilts of the dam and its environment were obtained. The Enguri Dam area is the object of large international projects: the Enguri Dam International Test Area (EDITA) is operating in the framework of the Open Partial Agreement on Major Disasters of the Council of Europe from 1995 [50,51]. The international projects DAMAST (2019–2022), ”Dams and Seismicity—Technologies for safe and efficient management of hydropower reservoirs“ and its successor “DAMAST-Transfer” aim to develop, install and test modern dam-monitoring systems as well as to apply nonlinear dynamics and machine learning methods for the analysis of observational data and predicting the safety of the system.

The hidden nonlinear structures in the time series of observations cannot be revealed with standard statistical methods; at the same time, the nonlinear analysis of these details can reflect significant deviations from the deformation pattern in the safe regime. In order to reveal such deviations, we used nonlinear dynamics methods [19,24,49,53], namely, mutual information (MI), recurrence plots (RP), recurrence quantification analysis (RQA), singular spectrum analysis (SSA) and Tsallis entropy (TE).

#### 4.3.2. The Process of Tilt/Strain Complexity Adaptation to Initial Reservoir Loading

The data on strain/tilt in the foundation of Enguri Dam reflect the change in the Earth’s crust’s dynamics from the natural state due to dam erection. In this section we consider the time intervals including those before dam erection as well as periods of dam erection and the initial irregular stage at first fillings.

**Tilts**. The data on tilts illustrate the evolution of corresponding crust tilt dynamics from the natural state, stages 1–2 (1974–1977), to the period of initial impoundment of the reservoir, stages 3–6 (1978–1982), and lastly, to the beginning of the regular water load/unload regime, stage 7 (1983–1985) [41]. Below we present the results of Tsallis entropy (Figure 7) and mutual information (Figure 8) calculation. It follows from these results that the extent of determinism decreases in the period of initial impoundment 1978–1982 (period of observation 4), when the natural ordering is destroyed due to initial reservoir impoundment.

The presence of an ordered structure in the tilt time series (Figure 7 and Figure 8) at intervals 1–2, i.e., before the dam construction; the decrease in regularity at intervals 3–6; and finally, the increase in order in period 7, when the regular load/unload regime began are evident. The data on strain/tilt in the foundation of Enguri Dam reflect the change in the Earth’s crust’s dynamics from the natural state due to dam erection. In this section we consider the time intervals including those before dam erection as well as periods of dam erection and the initial irregular stage at first fillings.

All applied methods document similar changes in tilt dynamics at the different stages of observation: namely, they indicate to the presence of an ordered structure in the tilt time series at intervals 1–2, i.e., before the dam construction; a decrease in regularity at intervals 3–6; and finally, an increase in order in period 7, when the regular load/unload regime began.

**Strains**. The branch fault of the main Ingirishi fault (Figure 6) crosses the foundation of Enguri Dam and as a real source of hazard should be monitored permanently. To control the fault behavior a quartz strainmeter was installed in the adit crossing the fault zone. In the following, we consider variation in the strain in the same main periods, which were singled out for tilts (Figure 9).

It is evident that the strain pattern (Figure 10 and Figure 11) mainly follows that of tilts, namely, there are different main periods of the strain dynamics: the initial period, (1974–1978), when the dynamics follows the natural pattern (without man-made impact); the transitional period of dam filling (1977–1982); and beginning of the accommodation of strain to the regular reservoir regime (1983–1985, Figure 10 and Figure 11).

The source of relatively strong ordering in all complexity measures in the natural state before the time when the dam building had begun (1971–1977) is not clear. We guess that in our case the seasonal regularity of strain can be connected with the snow load/unload, which can be quite significant, as the Enguri Dam is erected in a mountainous area. The period (1978–1982) when the initial ordering strongly decreases can be explained by the anthropogenic intervention (dam erection/initial filling), which in this stage is non-periodic. The last period, or stage 7 (1983–1985), is marked by increased ordering due to the beginning of the regular water load/unload regime (Figure 10 and Figure 11).

### 4.4. The Tilt/Strain Complexity in the Whole Period of Observations

#### 4.4.1. Strains in the Dam Foundation

Figure 12 presents the whole history of the fault zone extension (strain) (FZE) under the dam during almost a half-century, including the initial period discussed in the previous parts, which demonstrates that dramatic changes can happen in its dynamics with time. There are two main components of the fault strain rate α (SR) graph—the piecewise-linearly changing increment of the fault opening and superimposed on it quasi-periodic strain variation. The record of the fault extension linear component rate show that significant changes—namely, acceleration and deceleration—occur in 1984, 1987, 2004, 2013 and 2017. The quasiperiodic component of strain is due to the stabilized regime of water load/unload after 1984, which follows the transient period of initial reservoir filling analyzed in the previous section. The strain rate (SR) changes are due to the interplay of different decelerating/accelerating factors acting on the 10 m wide fault zone. The strongly fractured rock filling the zone material exhibits mechanical properties that range from fluid to solid behavior under varying stress. Such “phase transitions” are explained by the formation under applied stress of force chain networks, which ensure the ability of a granular system to support the applied stress. According to above model, the initial SR values of for 1974–1984 or $\alpha_{1}$ reflect the natural dynamics of the fault strain rate ($\alpha_{1}=230\;\mu m/\mathrm{year}$). This SR characterizes the mechanical properties of the natural system of force chains in the fault gorge under the natural strain rate. This natural strain rate decreased in the following 1984–2004 period ($\alpha_{2}=150\;\mu m/\mathrm{year}$), which could be caused by the damage of existing force chains by water load cycles as a result of a reversed-stress fatigue effect. There are also following epochs (2004–2013 and 2013–2017) in the strain rate history (Figure 12) with the second interchange of the linear components’ slope of the fault strain rate between $\alpha_{1}$ and $\alpha_{2}$, which we explain by the interplay of accelerating/decelerating epochs. In the most recent years (2018–2021) the strain rate decreased abruptly to $\alpha_{3}=44\;\mu m/\mathrm{year}$. A possible explanation of the strain rate deceleration is given in the following section.

We note also that there is a phase lag between the maxima of WL and the corresponding maxima in FZE variations: the fault zone displacement is retarded relative to the WL maxima by approximately 200 days. For the analysis of delay we use the equation of fluid diffusion front propagation in the fault over distance *r* in order to estimate the hydraulic diffusivity of the fault media between the reservoir and strainmeter location: *r* = (4*πDt*)^1/2^, where *r* is the distance between the perturbation source and a given point in the media and *t* is the time elapsed after the start of the pore pressure perturbation in the source. The distance *r* between the dam’s upper pool (pressure perturbation source) and the strainmeter site is 100 m and the time delay *t* between the WL and FZE maxima approximately 200 days. Substituting these data into the above diffusion equation, we obtain for the hydraulic diffusivity *D* of powdery rocks in the fault zone approximately $D\approx 5*10\;m^{2}/s$, which is not far from the typical results obtained in other studies.

#### 4.4.2. Nonlinear Dynamics of the Regular Dam Foundation Strains

The quasi-periodic strain variations superimposed on the piecewise-linear component of the Ingirishi branch fault (Figure 12) are the most interesting from the nonlinear dynamics point of view, as they can give information on the strain dynamics deviation from the regular (safe) regime. The fault crosses the foundation of Enguri Dam (Figure 6). We consider the fault strain data from 1974 to 2020 in order to study the impact of different factors on the strain dynamics using the methods of complexity theory. In Figure 13 we show the results of recurrence plot (RP) analysis of the fault displacements for 45 years of history before reservoir filling (1974–1978) and in the following several 5-year periods. Until 1987, there were no clear signs of recurrence in the RP plots of the strain. After the establishment of a regular WL regime (2014–2018), the repeating 5-year cycles appear in the RP graphs of both WL and fault displacement data. In the final 2015–2020 period the nonlinear dynamics analysis shows that the RP pattern became less regular. It should be noted that in the same period the fault strain rate α abruptly decreased to $\alpha_{1}=44\;\mu m/\mathrm{year}$; these data point to a significant change in the dynamics of the fault in the last years.

Analysis of other nonlinear characteristics (Figure 14) shows that they practically follow the pattern of %DET, of course except entropy. In general, all complexity characteristics follow the same pattern: in the first “natural” period of fault dynamics the determinism is relatively high, in the transitional period of reservoir filling the ordering is low, and in the following period the prolonged “regular” regime is established, which is replaced by a period of decreased regularity after 2012. Of course, the pattern of entropy is reversed: from 1985 to 2014, i.e., during the period of regular water regime, the entropy became negative: as we mentioned above, the entropy in the waiting time distribution after periodic forcing (periodic load/unload cycles of reservoir) decreased compared to the data before such forcing. The entropy decreased and even became negative as the system became less disordered. The above results demonstrate that the analysis of complexity in the monitoring data can be useful for the assessment of the stability of high dam functioning. For example, the monitoring data of the Enguri Dam document distinct deviations in the last years of the fault strain regime from the earlier pattern: the fault strain rate α abruptly decreased to $\alpha_{1}=44\;\mu m/\mathrm{year}\;$. In addition, at the same time determinism in the fault zone deformation time series decreased and entropy increased (Figure 14). The change in the strain regime in the last years can be connected either with the final stabilization of the fault motion or with only temporal braking of the fault motion due to a strong asperity. In the last case the stress will build up due fault motion retardation by asperity brake and finally can be dropped in a dynamic way at attaining the critical stress value. Consequently, the decrease in α as well as deviations in determinism and the entropy pattern in the last years’ data deserve more detailed study in the future.

#### 4.4.3. The Phenomenon of Reservoir-Induced Seismicity Synchronization (RISS)

The data presented in the previous sections show the impact of WL variations on the local shallow strain field; according to experimental data, even a small change in the reservoir load can activate local seismicity [42,43,54,56]. In this section we consider the influence of the WL change on the seismic regime of the area. The analysis shows that seismic events in the vicinity of the Enguri Dam area occur in the known seismic zones [57,58]. In the following, we show that this activity is associated with variation in the water level in the reservoir and is a typical example of reservoir-induced seismicity (RIS). As a rule, RIS is connected with a much deeper layer of the Earth’s core than the strain/tilt data considered in the previous part of the section. It is accepted that RIS is a transitional process connected with a process of filling the reservoir; the seismic activity increased (with a delay from years to days) after the water depth reached a value on the order of 100 m. After this adaptation period, the seismic rate decreases to almost initial values. At the same time, one important effect can be observed: the waiting times between EQs reveal a regular, close to periodic, time pattern. We name this effect Reservoir-Induced Seismicity Synchronization (RISS) [52,55].The effect seems to be doubtful, as the additional stress invoked by water level variation is very small—from several to hundreds of kilopascals—compared to the acting tectonic forces at the depth of several km, where the strain is on the order of 100 MPa [42,43]. At the same time, it is shown that exposure of a nonlinear system, such as an autonomous self-sustained oscillator, to the weak external forcing can affect its dynamics; as a result, even very weak external periodic forcing is able to adjust the natural rhythm of the system to the impact of applied forcing [13,14]. Figure 15 presents the graphs of the strain in the Enguri Dam foundation (black), water level in the lake (dark blue) as well as the quarterly number of weak EQs from October 2020 to February 2022, i.e., in the period when the WL regime of the dam was more or less regular. Note the two to three months’ retardation of seismic activity spikes relative to the abrupt seasonal change in the WL and strain [59].

Figure 16 presents the RQA and LZ long-term characteristics of earthquake waiting (interevent) times in 1974–2017 in the original seismic catalog in the Enguri Dam area within a 100 km radius for successive 24-month intervals. The %DET, Boltzmann entropy and trapping times (TT) analysis of the data show that, as in the case of strains (Figure 14), the regime of interevent (waiting times) also changes significantly during different periods of dam erection and exploitation. Particularly, in the initial period (1974–1986) there is a weak regularity in the waiting-time sequence of EQs, which is destroyed in the period of lake filling and following exploitation from 1986 to 1998. After this, from 2004 the regularity of interevent times increased strongly, almost to 90%.

According to the analysis of RQA and LZ data (Figure 16), the regularity of the seismic process increased when the load/unload of the reservoir became periodical, but with a considerable delay. Besides, the decrease in entropy is not as dramatic as in case of strain (Figure 14), as the data of the seismic catalog were collected in a region with a radius R = 100 km from the dam. Of course, this means that the local dam effect is mixed with regional seismic events, which makes the influence of the former less obvious. The same results were obtained in the analysis of waiting times using Mahalanobis distance analysis [60]. It is interesting that in the time patterns of the fault strain rate (Figure 14) and weak seismicity (Figure 16) we can mark out three different periods: initial (constant strain rate of the fault and weak order in the seismic regime), transitional during initial lake filling (the increase in disorder in the seismic regime) and stable (close to periodic change in strain rate and high ordering of EQ waiting times). At the same time, the beginning of the regular period in the waiting time distribution of EQs lags considerably behind the strain regime pattern: regularity in the strain time series increases strongly after 1984 (Figure 12), but the EQ waiting times became regular only after 2004–2005, which means that the seismicity regime ordering by the periodic water load (1998–2004) lags behind that of the strain regularization period (1980–1982) by approximately 20 years, i.e., the adaptation of the seismic regime to periodic loading takes much more time than that for deformation.

## 5. The Effect of Weak Forcing of Different Origin on the Dynamics of Seismic Events

In this section we present the examples of EQ synchronization by relatively weak natural and man-made processes, which quite often were rejected due to the great difference between the energy of the impact and outcome. As we showed earlier, such an effect is explained by a nonlinear response of the system to a weak periodic forcing [13]. Examples of such strong synchronization effects caused by weak forcing in the laboratory and nature are plenty: besides the reservoir periodic loading considered above [32,52,55], synchronization of EQs can be due to the impact of strong electromagnetic pulses, Earth tides and teleseismic waves from strong remote EQs [50,60,61,62].

The effect of strong electromagnetic pulses. From 1975 to 1996 near Bishkek polygon (Central Asia), experiments were carried out to study the effect of strong electromagnetic pulses generated by the MHD with some regular time intervals on the seismic regime of the region [63] (pp. 324–327). RQA analysis of the data documents the determinism (% DET) as well as laminarity (%LAM) calculated for the EQ time series, namely for experimental time intervals between seismic events in the original catalog and the same data after chaotic shuffling of MHD firings times (Figure 17). It is evident that EQ waiting times are much more regular in the intervals of MHD impact, i.e., in the time windows from 5 to 13; the regularity disappears after the shuffling of waiting times and the %DET/%LAM return to the background pattern for the region.

The Earth/oceanic tides and seismicity. The conclusions of related publications are very different: in [62,63,64,65,66] a significant correlation between tides and seismicity was established, though [67] rejects the possibility of such an effect, arguing that the tectonic forces needed for EQ generation are many orders larger than tidal stress.

Generally, from the point of view of complexity theory even a weak periodic impact such as tides can synchronize seismic activity due to the accumulation-and-fire effect, considered in synchronization theory [14]; according to it, the synchronization of seismic processes by a weak forcing is possible in the case where the EQ nucleation time $T_{0}$ is close to the period of forcing $T:\left(T-T_{0}\right)\approx 0$; this effect is called 1:1 synchronization. This condition points to the conclusion that tidal synchronization of seismicity should be expected for fast M2, O2 tidal components and seismic/seismo-acoustic events with a short nucleation time [14]. Of course, besides 1:1 synchronization, the high-order synchronization can also take place when the EQ nucleation time $T_{0}$ is larger/smaller than the forcing period T. This can occur at the multiples *n:m* (*n* forcing pulses within *m* cycles of nucleation time), when many Arnold’s tongues can be formed [14].

Nonvolcanic tremors. It has been discovered recently that teleseismic surface waves of remote strong earthquakes can synchronize a new class of tremors—non-volcanic tremors (NTV) [64]. The similar effect of synchronization was observed in laboratory conditions: periodically forced stick–slip (with a forcing period *T*) also generated weak acoustic emission signals at multiples of T (Figure 10.8 and Figure 10.9 in [63]).

## 6. Conclusions

The laboratory experiments carried out on the mechanical and electromagnetic periodic forcing of stick–slip processes in a slider–spring system can be considered as modeling the reaction of big engineering systems such as dams to periodic loading. We show that even a weak external forcing leads to phase synchronization of the stick–slip process due to the nonlinear dynamic nature of the process. We show also that besides the 1:1 synchronization mode, such forcing also invokes the high-order synchronization on the order of *n:m*. The experiments with varying intensity/frequency of forcing allow delineation of the synchronization regions in the phase space plots. These results show that that forced stick–slip is an example of the general class of integrate-and-fire systems with an inverse bell-curve form of the phase space plot of synchronization area (so-called Arnold’s tongues).

After laboratory modeling, we apply the above approach of nonlinear dynamics to the analysis of natural systems under weak periodic forcing, namely, to the reaction of strain/seismicity time series recorded near large dams to water level (WL) variation the in reservoir. The complexity analysis of these data can reveal deviations of the tilts/strains/seismic regime from the regular stable pattern. The time series of fault zone extension (FZE), or extension of the fault zone crossing the dam foundation, is marked by a high determinism in the period of regular WL variation: the strain varies in a quasiperiodic manner according to a seasonal water level variation, with a retardation by 200 days behind the WL extrema. This time lag, due to the diffusion of water from the lake to the strainmeter, crossing the fault zone (FZ), allows us to calculate the hydraulic diffusivity D of rocks in the fault zone as $D\approx 5*10\;m^{2}/s$.

There are several different stages in the nonlinear dynamics of both strain and seismicity time series near large dams: i. close to regular before dam erection; ii. irregular during dam construction and initial filling of the lake; iii. regular during periodic loading/unloading of the reservoir with characteristic nonlinear dynamics; iv. the last four-year period with a reduced regularity and decreased long-term strain rate component. The beginning of the ordering process in the seismic regime of the dam area due to periodic charge–recharge of reservoir is retarded by 20 years relative to the fault strain synchronization by the water load.

The analysis of the dynamics of strains/tilts/seismicity time series of Enguri Dam and the surrounding area allows us to establish the pattern of nonlinear dynamics in the normal (safe) regime, as well as detect the significant deviations from it. The analysis of FZ dynamics can help decision makers to judge whether the dynamics of observed data is regular (safe mode for a given construction) or there are anomalies in the dynamics, possibly pointing to developing dangerous processes.

## Figures and Tables

**Figure 1 entropy-25-00467-f001:**
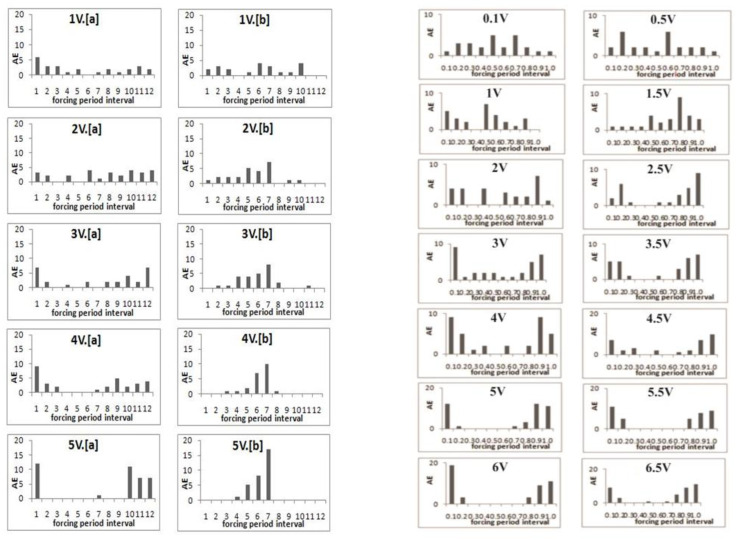
Distribution of acoustic emission onsets relative to mechanical forcing period phases (in decimals) for different intensities of normal (the left two columns) or tangential (the right two columns) forcing. In the both tangential and normal forcing experiments the left column shows results for onsets and the right one for terminations of mechanical forcing signals. Forcing frequency at normal forcing is 20 Hz and at tangential forcing, 80 Hz. In all above figures *V* is used to show the applied voltage [45].

**Figure 2 entropy-25-00467-f002:**
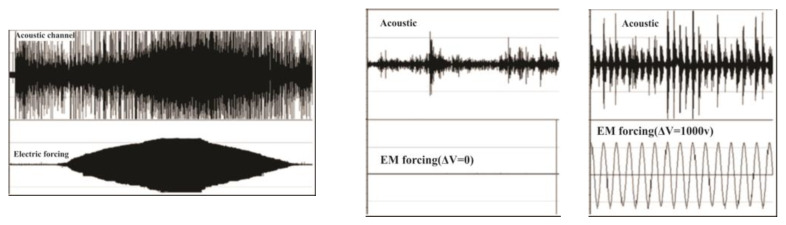
Acoustic emission during slip under variable from zero to 1000 V external periodical voltage (**left**); the extended part of record at zero EM forcing (**middle**); the extended middle part of record under maximal EM forcing (**right**); note complete phase synchronization. There are two sections in the graphs: the upper one shows slip-generated AE events and the lower one, applied AE hits [45].

**Figure 3 entropy-25-00467-f003:**
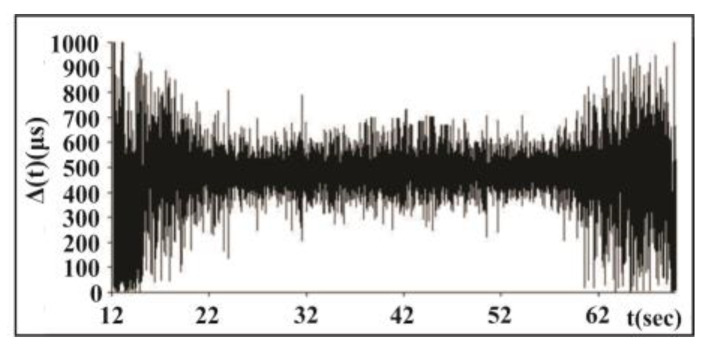
Time series of waiting times between consecutive amplitudes of acoustic signals in consecutive Δt periods of external forcing for a whole record. Significant EM forcing was applied in the time interval from 22 to 60 s of the stick–slip process [45].

**Figure 4 entropy-25-00467-f004:**
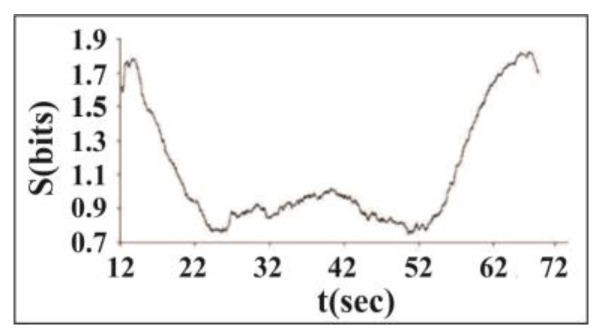
Variation of Shannon entropy *S* of phase differences calculated for consecutive sliding windows containing 500 events. Synchronization is strong in the area from 22 to 60 s or at forcing intensities from 500 to 1000 V [45].

**Figure 5 entropy-25-00467-f005:**
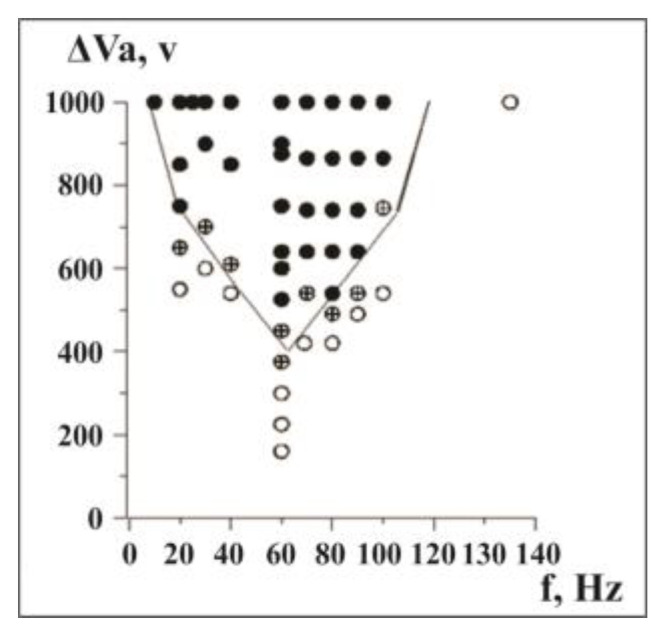
Stick–slip synchronization area (Arnold’s tongue) for various intensities $V_{a}$ and frequencies (*f*) of the external EM forcing. Filled circles—perfect, circles with crosses—intermittent and empty circles—absence of synchronization [45].

**Figure 6 entropy-25-00467-f006:**
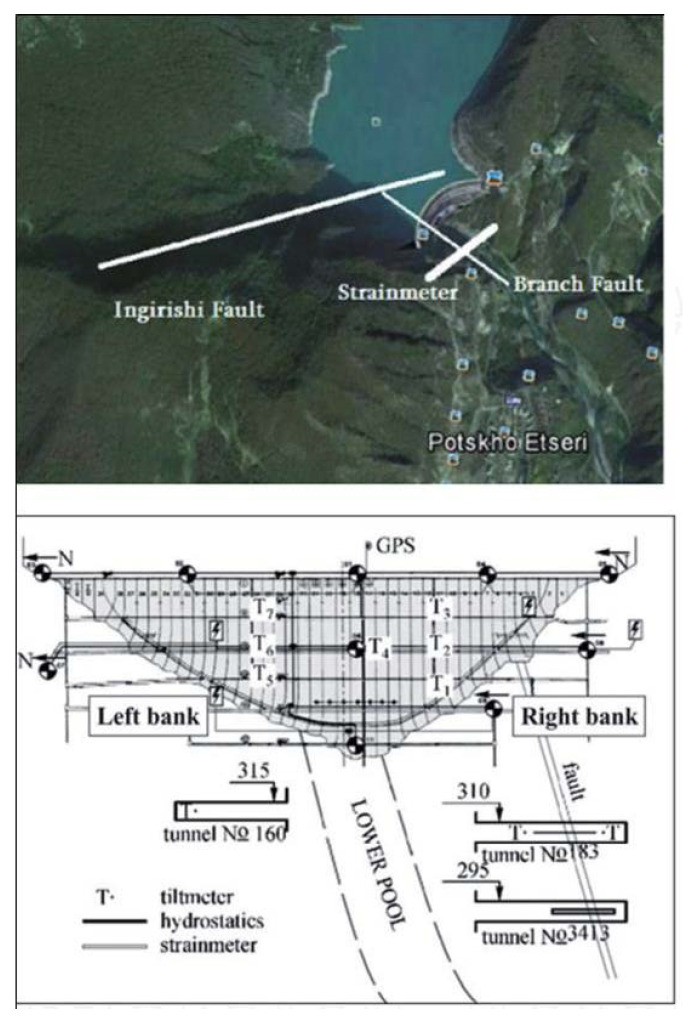
Enguri Dam area from space (**upper**) and a cross-section of Enguri Dam with a monitoring system viewed from the upper pool (**lower**). T—Tiltmeters; S—Laser strainmeter [52].

**Figure 7 entropy-25-00467-f007:**
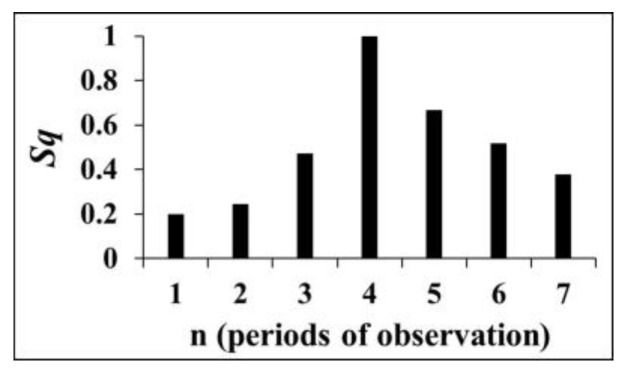
Tsallis entropy values of Earth tilt data series; for demonstrative purposes, here we present results for *q* = 0.5 [54].

**Figure 8 entropy-25-00467-f008:**
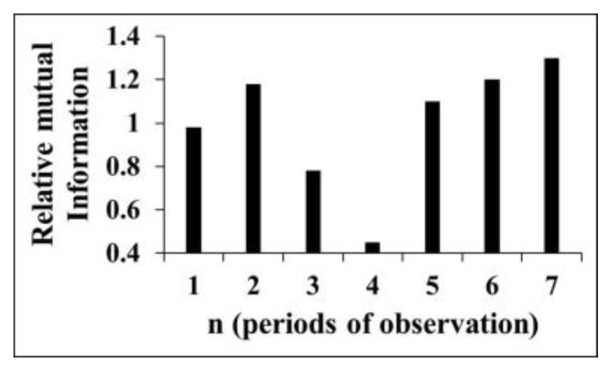
Relative mutual information values. Numbers in abscissa correspond to periods of observation [54].

**Figure 9 entropy-25-00467-f009:**
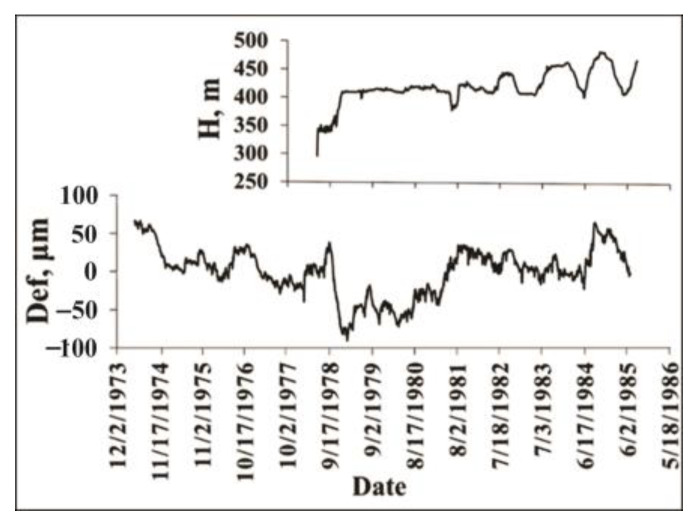
Variation in water level in the lake (H, m) and deformation on the fault (Def, µm) in the period 1974–1985 [55].

**Figure 10 entropy-25-00467-f010:**
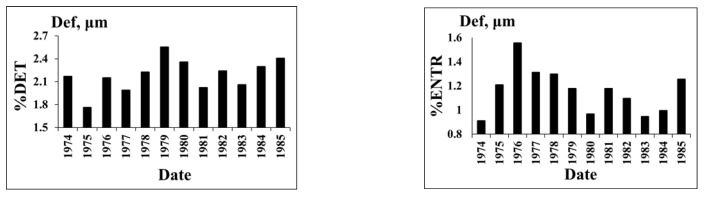
RQA analysis of deformation %DET and Boltzmann entropy (BE) for the initial period 1974–1985.

**Figure 11 entropy-25-00467-f011:**
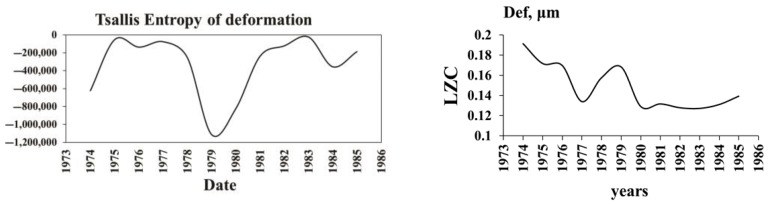
Tsallis entropy and Lempel–Ziv complexity of deformation for the initial period 1974–1985.

**Figure 12 entropy-25-00467-f012:**
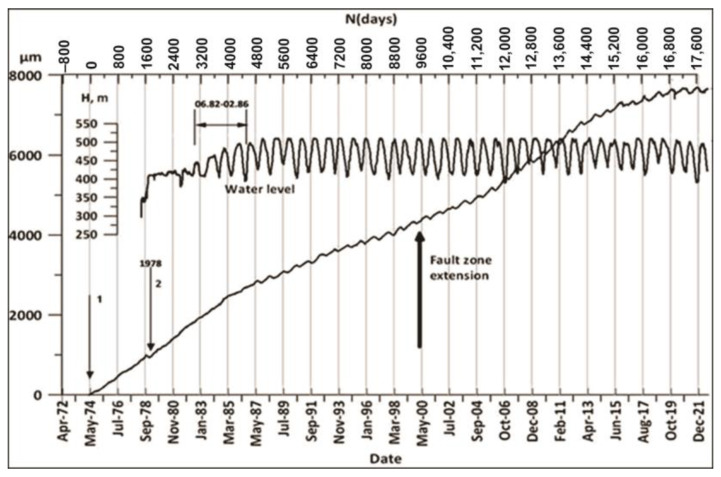
The full-time dynamics of WL in the Enguri Dam reservoir (meters) from 1978 to 2021 (upper curve) and extension/compaction (in micrometers) of the Ingirishi branch fault crossing the foundation of the dam from 1974 to 2021 (lower curve). Arrow 1 marks the start of strainmeter monitoring in 1974 and arrow 2, the beginning of reservoir impoundment [55].

**Figure 13 entropy-25-00467-f013:**
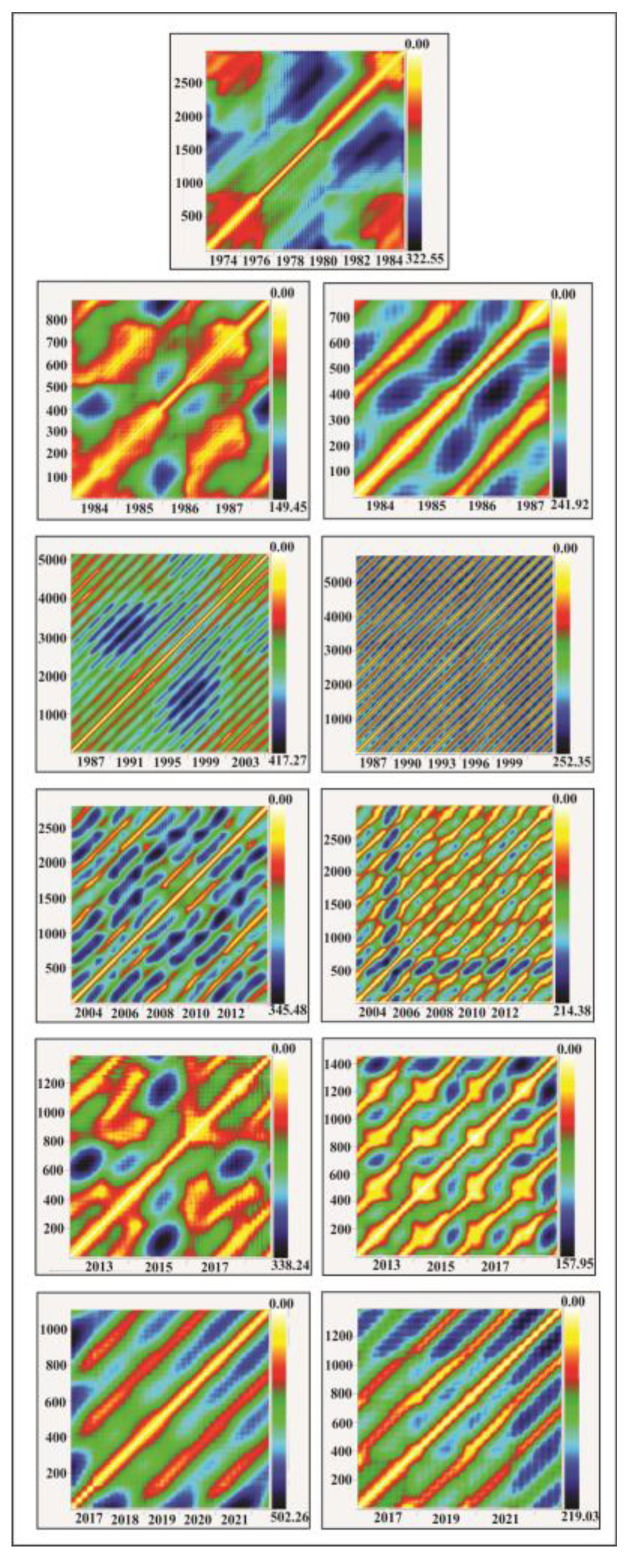
Recurrence plot (RP) of fault displacements’ “time story“ for several successive periods (left column) as well as RP for the water level regime (right column). The first figure is the RP of fault displacement before reservoir filling (1974–1978), i.e., the RP for the natural state; the next figures, P during regular reservoir exploitation. Identical yearly cycles are present in the RPs of fault displacement, beginning from the 1987–2013 period; the last RPs beginning from 2015 differ from the previous regular patterns due to a less-regular WL regime (Figure 12).

**Figure 14 entropy-25-00467-f014:**
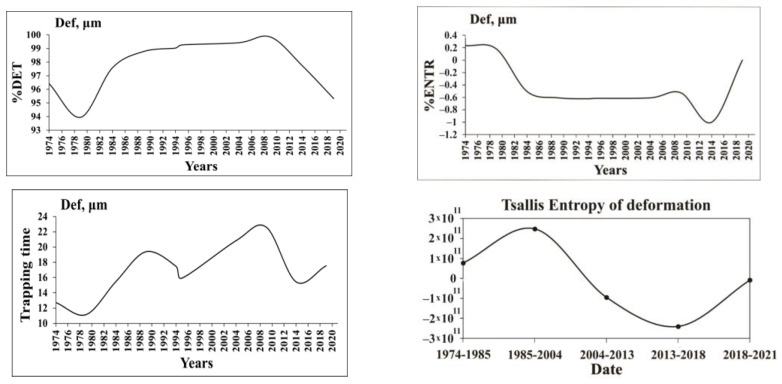
The graphs illustrate the nonlinear dynamics of 45 years’ fault displacement time series for the characteristics RQA determinism (%DET), Boltzmann entropy (%ENTR) and Tsallis entropy (TE) for the order 4 of dam deformation, Trapping times (TT) for period 1974–2021 [55].

**Figure 15 entropy-25-00467-f015:**
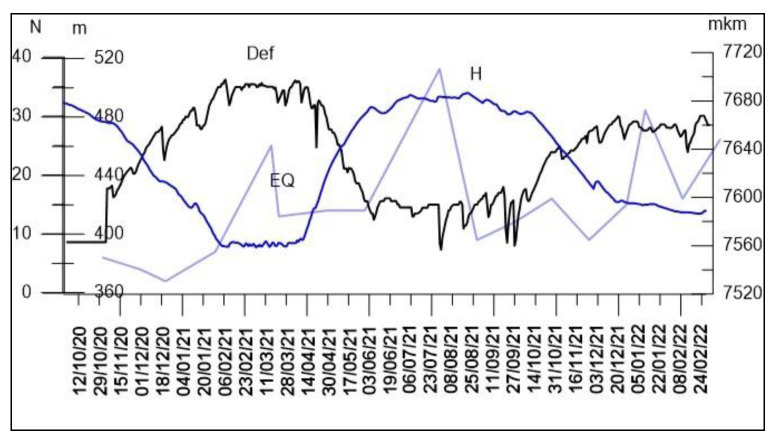
Graphs of the strain in the Enguri Dam foundation (black), water level in the lake (dark blue) and the quarterly number of EQs from October 2020 to February 2022. Note retardation of seismic activity relative to abrupt seasonal change in the WL [59].

**Figure 16 entropy-25-00467-f016:**
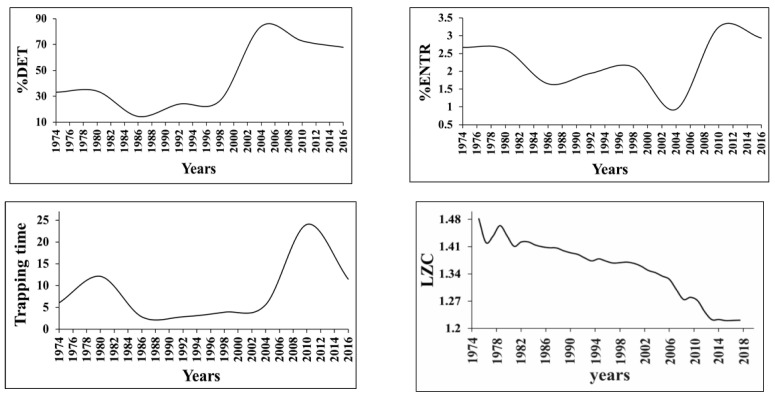
Earthquake waiting (interevent) time characteristics—%DET, trapping times (TT), Boltzmann entropy and LZ complexity—of the original seismic catalog (1974–2017) in the Enguri Dam area within a 100 km radius [55].

**Figure 17 entropy-25-00467-f017:**
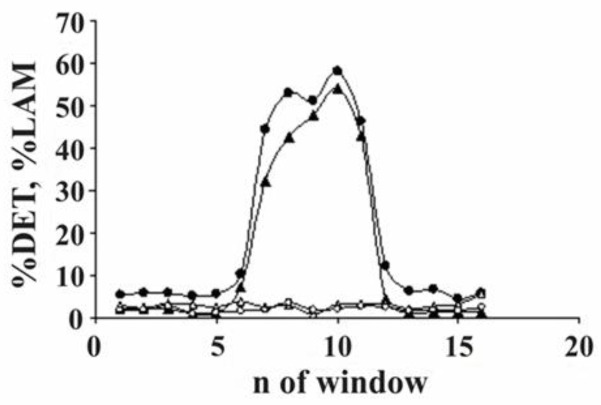
The determinism % DET (circles) and laminarity %LAM (triangles) of the waiting times in the seismic time series of the Bishkek polygon at a representative threshold M > 2.5: the strong MHD pulses were applied in time windows 5–13. The filled symbols correspond to the experimental data and void ones to the same data after shuffling of waiting times [63].

## Data Availability

No new data were created of analyzed in this study. Data sharing is not applicable to this article.

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
