# Peer review of "Complexity in Geophysical Time Series of Strain/Fracture at Laboratory and Large Dam Scales: Review"

_entropy, 2023, doi:10.3390/e25030467_

Round 1

Reviewer 1 Report

In this manuscript (ms), the authors present a study of various complexity measures in geophysical time series (Earth tilt, deformation, fault displacement, etc) recorded close to the Enguri dam together with similar studies in laboratory experiments.  The scope of the ms, which focuses on reviewing the subject, is to show that the obtained data on the dynamics of strain/seismicity near large dams can be used for assessment of the possible risks in order to help decision makers to judge whether the dynamics of observed data is regular or there are anomalies in these dynamics, possibly pointing to developing dangerous processes. The ms succeeds in its scope and hence it deserves publication in Entropy.

The presentation, however, needs significant improvement before publication. The main points that the authors should improve when resubmitting are the following:

1)The Reference list should also include works that have not been mentioned although related to the topics discussed. Moreover, typing errors lead to citations to papers that are irrelevant to the topics discussed.

1A)In lines 54-56, the authors mention the presence of long-term correlations in the fracture of solids and seismic processes. The following recent publications discuss long-term correlations in seismic processes:

N.V. Sarlis et al., "Micro-scale, mid-scale, and macro-scale in global seismicity identified by empirical mode decomposition and their multifractal characteristics”, Scientific Reports 8 (2018), 9206, https://doi.org/10.1038/s41598-018-27567-y

N.V. Sarlis et al., "Investigation of the temporal correlations between earthquake magnitudes before the Mexico M8.2 earthquake on 7 September 2017", Physica A 517 (2019) 475-483, https://doi.org/10.1016/j.physa.2018.11.041

E. S. Skordas et al., "Detrended fluctuation analysis of seismicity and order parameter fluctuations before the M7.1 Ridgecrest earthquake", Natural Hazards 100 (2020), 697-711, https://doi.org/10.1007/s11069-019-03834-7

P. K. Varotsos et al., "Estimating the Epicenter of an Impending Strong Earthquake by Combining the Seismicity Order Parameter Variability Analysis with Earthquake Networks and Nowcasting: Application in the Eastern Mediterranean", Applied Sciences 11 (2021) 10093, https://doi.org/10.3390/app112110093

Hence, some of them should be listed together with Reference “[8]” in line 56.

1B)In lines 74-79, several methods for the quantitative analysis of synchronization are mentioned. The following recent publication:

N.D. Tsigkri-DeSmedt et al., “Shooting solitaries due to small-world connectivity in Leaky Integrate-and-Fire networks”, Chaos 31 (2021) 083129, https://doi.org/10.1063/5.0055163

presents a method (local order parameter) that hasn’t been mentioned in the above and hence it should be added in the list for the readers’ better information.

1C)In lines 104-108, various methods for quantitative analysis of complex systems are mentioned. The authors do not mention natural time analysis, e.g.,

P.A. Varotsos et al. “Perspective: Self-organized Criticality and Earthquake Predictability: A long standing question in the light of natural time analysis”, EPL 132 (2020) 29001, https://doi.org/10.1209/0295-5075/132/29001

P.A. Varotsos et al., “Order Parameter and Entropy of Seismicity in Natural Time before Major Earthquakes: Recent Results”, Geosciences 12 (2022) 225, https://doi.org/10.3390/geosciences12060225

which has been applied in a variety of complex systems, see, e.g.,

P.A. Varotsos et al., “Natural Time Analysis: The new view of time. Precursory Seismic Electric Signals, Earthquakes and other Complex Time-Series” (Springer-Verlag, Berlin Heidelberg) 2011 https://doi.org/10.1007/978-3-642-16449-1

This method should be also mentioned in lines 104-108.

                1D)In line 114, a citation to the Detrended Fluctuation Analysis is expected but Reference [12], mentioned there, is irrelevant. The authors should at least mention the following References

C.-K. Peng et al., “Finite-size effects on long-range correlations: Implications for analyzing DNA sequences”, Phys. Rev. E  47(1993) 3730–3733, https://doi.org/10.1103/PhysRevE.47.3730

C.-K. Peng et al., “Mosaic organization of DNA nucleotides”, Phys. Rev. E 49 (1994) 1685–1689, https://doi.org/10.1103/PhysRevE.49.1685

in which DFA was introduced.

                1E)In lines 126-128, the authors discuss the importance of Tsallis entropy in the analysis of laboratory fracture experiments and seismic processes. However, References [14, 15, 19], which are mentioned there, are irrelevant. The following recent References might be useful on this subject:

A. Loukidis et al. “Non-Extensive Statistical Analysis of Acoustic Emissions Recorded in Marble and Cement Mortar Specimens Under Mechanical Load Until Fracture” Entropy 22 (2020) 1115, https://doi.org/10.3390/e22101115

E. S. Skordas et al. "Precursory variations of Tsallis non-extensive statistical mechanics entropic index associated with the M9 Tohoku earthquake in 2011", European Physical Journal – Special Topics 229 (2020) 851-859, https://doi.org/10.1140/epjst/e2020-900218-x

A. Posadas and O. Sotolongo-Costa, “Non-extensive entropy and fragment–asperity interaction model for earthquakes”, Communications in Nonlinear Science and Numerical Simulation 117 (2023) 106906, https://doi.org/10.1016/j.cnsns.2022.106906

                1F)The authors should also check the relevance of References [21,22] to machine learning as mentioned in the context of lines 129-131 and probably replace them.

                1G)In lines 152-155, the authors discuss precursory to fracture acoustic emission (AE) phenomena. In addition to Reference [29], the following more recent References

A.Loukidis et al., “Fracture analysis of typical construction materials in natural time”, Physica A 547 (2020) 123831. https://doi.org/10.1016/j.physa.2019.123831

A.Loukidis et al., "Natural time analysis of acoustic emissions before fracture: Results compatible with the Bak-Tang-Wiesenfeld model” EPL 139 (2022) 12004, https://doi.org/10.1209/0295-5075/ac7bee

should be also mentioned for the readers’ better information.

                1H)In lines 528-534, the very interesting experiments performed at the Bishkek Research Station of the Russian Academy of Sciences (former USSR Academy of Sciences) are mentioned but the cited there Reference [48] discusses results from California. Moreover, Figures 10.28 and 10.29 do not exist in Reference [48]. The authors should replace [48] with the appropriate one they want to use.

 2)The Methods section should include all the definitions of the quantities used in the ms and especially in its Figures so that it is self-contained.

3)Tsallis entopy is positive, but it appears negative in Figures 11 and 14.

4)Several figures need significant improvement. For example, I cannot follow the symbols and notation in Figure1. V or v means Volt? What does forcing period interval means? AE in the vertical axes is AE events or AE hits? The quantities presented in Figures 7 and 8 should be also briefly defined in the Methods section. Especially in Figure 8, I cannot understand the 8th period that appears, an explanation should be added in the figure caption. Figure 13 is unreadable, better quality is needed.

5)Various typing mistakes should be corrected, see, e.g., \omega_0 in lines 72 and 73, 10^9-10^12 in line 88, it changes in line 169, means in line 215, F_pi is explained twice in line 228, 10^9 m^3 in line 284, MHD in line 530 is not defined, FZE is defined twice see lines 381 and 589, etc.

6)The statement “D of rocks in the zone as: ?≈510?^2/?” that appears in the conclusions is not supported by the presented results.

In view of the above, I consider that the ms needs  revision along the lines mentioned above. In view of the importance of the scope of the present ms, I would be glad to suggest publication of an appropriately revised ms.

Author Response

Thank you for your kind review. Please see the attached file.

Reviewer 2 Report

the file in the attachment

Author Response

Thank you for the kind review. Please see the attached file.
